# Biomolecules, Fatty Acids, Meat Quality, and Growth Performance of Slow-Growing Chickens in an Organic Raising System

**DOI:** 10.3390/ani12050570

**Published:** 2022-02-24

**Authors:** Wittawat Molee, Wichuta Khosinklang, Pramkamon Tongduang, Kanjana Thumanu, Jirawat Yongsawatdigul, Amonrat Molee

**Affiliations:** 1School of Animal Technology and Innovation, Institute of Agricultural Technology, Suranaree University of Technology, Nakhon Ratchasima 30000, Thailand; wichutaploy085411@gmail.com (W.K.); woonwoon.kati@gmail.com (P.T.); amonrat@sut.ac.th (A.M.); 2Research and Facility Department, Synchrotron Light Research Institute (Public Organization), Nakhon Ratchasima 30000, Thailand; kthumanu@gmail.com; 3School of Food Technology, Institute of Agricultural Technology, Suranaree University of Technology, Nakhon Ratchasima 30000, Thailand; jirawat@sut.ac.th

**Keywords:** organic raising system, meat quality, omega-3 fatty acid, slow-growing chicken, synchrotron FTIR

## Abstract

**Simple Summary:**

The increasing demand for nutritionally rich quality products by health-conscious consumers has raised the need to explore alternative farming systems such as organic farming. In this study, we report the efficiency of Korat chickens grown in organic farm conditions. The study demonstrates not only that the slow-growing Korat chicken is suitable for organic farming, but also that the organic raising system improves its growth performance and meat quality. Furthermore, the study unveils a set of biochemical traits using synchrotron radiation-based Fourier transform infrared that show significant differences between the meat quality of chickens raised under conventional and organic raising systems, suggesting their potential use as markers to monitor the meat quality. The findings of this study provide evidence for the potential of organic raising systems for commercial adoption in tropical areas such as Southeast Asia.

**Abstract:**

This study was to determine the effect of the organic raising system (OR) on growth performance, meat quality, and physicochemical properties of slow-growing chickens. Three hundred and sixty (one-day-old) Korat chickens (KRC) were randomly assigned to control (CO) and OR groups. The groups comprised six replicates of thirty chickens each. The chickens were housed in indoor pens (5 birds/m^2^), wherein those in OR had free access to Ruzi pasture (1 bird/4 m^2^) from d 21 to d 84 of age. In the CO group, chickens were fed with a mixed feed derived from commercial feedstuffs, while those in the OR group were fed with mixed feed derived from organic feedstuffs. The results revealed a lower feed intake (*p* < 0.0001) and feed conversion ratio (*p* = 0.004) in the OR. The OR increased total collagen, protein, shear force, color of skin and meat, and decreased abdominal fat (*p* < 0.05). The OR improved fatty acid with increased DHA, n-3 PUFA, and decreased the ratio of n-6 to n-3 PUFA in KRC meat (*p* < 0.05). The synchrotron radiation-based Fourier transform infrared spectroscopy and correlation loading analyses confirmed these results. In conclusion, our results proved that OR could improve growth performance and meat quality and suggested the raising system be adopted commercially. In addition, the observed differences in biochemical molecules could also serve as markers for monitoring meat quality.

## 1. Introduction

Raising systems have become a more serious issue, particularly in terms of animal welfare [1]. The strategies to increase the production rate to meet the growing demand for chicken meat have led to unintentional negative effects such as muscle abnormalities and increased susceptibility to stress-induced myopathy in chickens [2]. However, the increasing health consciousness amongst consumers has increased the interest in environmentally friendly animal products produced on natural or organic farms.

The organic raising system (OR) is a poultry management system where the birds are fed only with organic feed (produced without chemical fertilizers and pesticides) and allowed to grow and express their natural behaviors without the use of chemicals such as antibiotics and other drugs [3]. All ORs are free-range systems, as they allow free outdoor access; however, the reverse is not true, as the free-range systems that are not OR use general feed, medications, and chemicals [4]. The OR has gained increasing interest worldwide because it is environmentally friendly, its products are free from chemical residues, and it follows a high standard for the welfare of birds [3]. Chickens raised in OR have a high nutritional value in terms of protein, total collagen, and total omega-3 polyunsaturated fatty acid (n-3 PUFA) contents in meat [1,5,6,7]. However, the breeds suitable for OR need to be tolerant, resilient, adaptable, capable of utilizing quality balanced feed [8,9,10], and incur a low cost of production. Therefore, not all breeds are adaptable, but the slow-growing chickens seem to be suitable for the OR. The Korat chicken (KRC) is one such slow-growing chicken with an average daily gain (ADG) of 19.8 to 21.0 g/d, and it takes about 70 d to reach the marketable body weight (BW) of ~1.2 kg [11]. It is a hybrid chicken obtained from the cross of Suranaree University of Technology (SUT) as a dam line and the Thai native chickens (Leung Hang Khao, LHK) as a sire line. KRC is recognized for its meat quality and holds promise of being an efficient occupation for Thai smallholder farmers as well as smallholder farmers across Southeast Asia in the near future. However, it has not gained the expected levels of scientific interventions. Furthermore, though Southeast Asia is one of the largest chicken-producing regions, the efficiency of OR farming in Southeast Asia has not been explored fully. Therefore, to unveil the efficiency of OR for tropical areas, particularly Southeast Asia, it is essential to investigate the effects of OR on the growth performance and meat quality of slow-growing chickens compared to the conventional raising system.

Several studies have investigated the effect of OR on the growth performance and meat quality [12,13,14,15] of chickens; however, they have reported inconsistent results attributable to the differences in the breed, feed, and experimental sites used by them. Moreover, fatty acids (FA), particularly n-3 PUFA such as docosahexaenoic acid (DHA) and eicosapentaenoic acid (EPA), are expected to be higher in the meat of chickens that were raised in OR, because organic chickens can get α-linolenic acid (ALA) from pasture, which is the precursor of n-3 PUFA [14]. Nevertheless, some studies have reported contradictory results for the effects of OR on the fatty acid profiles of meat [16]. 

The raising system affected lipid oxidation [17], resulting in a change in secondary protein structure (such as α-helix and β-sheet) [18], and may alter the biochemical compositions in meat, or the contents of glycogen, which have been shown to play major roles in determining meat quality [19,20,21]. Therefore, it is expected that monitoring the changes in the biochemical composition of meat with high sensitivity could help understand and explain the effect of the raising system on meat quality. Fourier transform infrared microscopy is a powerful technique used for biological analysis and monitoring the changes in biochemical compositions at the molecular level [22]. However, the global light source used for this technique does not have enough power to penetrate the cell and a lower ability to detect changes compared with light sources from synchrotrons radiation [23]. 

Synchrotron Radiation-Fourier Transform Infrared (SR-FTIR) spectroscopy is a highly sensitive and powerful technique because it is extremely intense (hundreds of thousands of times more intense than that from conventional X-ray tubes) and highly collimated [24]. In addition, this technique is fast, inexpensive, and non-destructive compared to conventional methods. It can provide unique information and a high performance to detect biochemical compounds at the molecular level, such as proteins, lipids, and glycogen [25]. Moreover, the efficiency of FTIR spectroscopy to investigate the change in biochemical composition in KRC meat is evident from previous studies. For instance, Poompramun et al. [26] successfully evaluated the differences in the biochemical composition of KRC thigh meat from the high and low feed conversion ratio (FCR) groups. We hypothesized that SR-FTIR could help monitor the quality traits in meat obtained from different raising systems.

The present study investigates the effects of OR on the growth performance, physicochemical properties, and biochemical composition of meat using SR-FTIR in KRC. Furthermore, in this study, KRC was used—mainly to identify its potential as one of the representatives of slow-growing chickens in OR and represent itself.

## 2. Materials and Methods

### 2.1. Ethics Statement

In the present study, all procedures were approved by the Ethics Committee on Animal Use of SUT, Nakhon Ratchasima, Thailand (user application ID: U1-02633-2559).

### 2.2. Birds, Experimental Design, and Diets

This study was conducted from January to April 2018. The experimental site was located at the coordinates latitude 14°53′13″ N and longitude 101°59′42″ E. The temperatures varied from 20.0–35.5 °C with an average relative humidity of 76% (Nakhon Ratchasima Meteorological Department, Nakhon Ratchasima, Thailand). Before this study, heavy metal levels in water and soil were assessed, and the experimental area was not treated with pesticides or herbicides. All animals were raised according to the National Bureau of Agricultural Commodity and Food Standards [27]. 

Three hundred and sixty (1-day-old) mixed-sex Thai native crossbred chicks, KRC, were produced at the Poultry Research Unit of the University Farm. The chicks were vaccinated against Marek’s disease on d 1, Newcastle disease and infectious bronchitis on d 7 and 21, and Gumboro disease on d 14. After hatching, the chicks were randomly allocated to two different raising systems (considered treatment groups) using a completely randomized design—each treatment group comprised 6 pens and 30 chickens per pen. In the CO (control group), chickens were housed in an indoor pen (5 birds/m^2^), fed with mixed feed derived from commercial feedstuffs, while the chickens in the OR group were fed with mixed feed derived from certificated organic feedstuffs in an indoor pen (5 birds/m^2^) with free access to an outdoor Ruzi pasture (4 m^2^/bird) from d 21 of age to slaughter age (d 84). Ruzi pasture, planted from seed and grown by irrigation, is very palatable and tolerates moderately heavy grazing, and the chickens were allowed to eat this grass daily. The organic feed content in the starter, grower, and finisher diets fed to OR chickens were 96.20%, 96.65%, and 96.85%, respectively. The experimental diets used in both raising systems are shown in Table 1—the energy and protein levels of the diets were adjusted to the same level. The chickens had ad libitum access to feed and water throughout the experimental period.

### 2.3. Measurements and Chemical Analyses

#### 2.3.1. Growth Performance and Carcass Composition

Growth performance was estimated by assessing the BW and feed intake (FI) every week, and subsequently, BW gain (BWG) and FCR were calculated. The percentages of eviscerated carcasses and abdominal fat were measured as a ratio of the live chicken’s BW after feed withdrawal. The percentages of the breast, thigh, and drumstick were estimated as the percentages of the chilled carcass weight.

#### 2.3.2. Sample Collection

At slaughter age (d 84), 24 chickens from each group were randomly selected and electrically stunned, and their feathers were removed with a machine. Then, they were scalded and eviscerated manually. The carcass composition and meat quality were measured in 12 chickens per treatment (6 males and 6 females). The proximate composition, FA profile, cholesterol content, and nucleotide content were estimated from the breast and thigh meat samples obtained from the remaining 12 chickens (6 males and 6 females). The samples were stored at −20 °C till analyses.

#### 2.3.3. Drip Loss Measurement

The breast and thigh meat samples were cut into 1.5 cm (width) × 3.0 cm (length) × 0.5 cm (thickness) pieces from the same position after chilling for 24 h. Then, the cut meat samples were hung inside a chilled storage room at 4 °C for 24 h. The drip loss was estimated by using the following formula:$$Drip\;Loss\;\left(\%\right)=\frac{\left(Weight_{before\;storage}-Weight_{after\;storage}\right)}{Weight_{before\;storage}}\times 100$$

#### 2.3.4. Cooking Loss Measurement

After thawing the breast and thigh samples overnight, they were weighed and boiled in a water bath in open plastic bags until an internal temperature of 80 °C was reached. Cooking loss was calculated as follows:$$Cooking\;Loss\;\left(\%\right)=\frac{\left(Weight_{before\;boiling}-Weight_{after\;boiling}\right)}{Weight_{before\;boiling}}\times 100$$

#### 2.3.5. Warner-Bratzler Shear Force Measurement

A texture analyzer (TA-XT2, Texture Technologies Corp., Scarsdale, NY, USA) was used to determine the shear force of the cooked breast and thigh samples. At least two subsamples of 2.0 cm (width) × 3.0 cm (thickness) × 0.5 cm (length) were cut parallel to the muscle fibers. The crosshead speed was set at 20 cm/min, and the shear force was calculated following the method described by Wattanachant et al. [28].

#### 2.3.6. Morphological Analysis

Tissue samples were fixed in 10% formalin solution for 24 h at room temperature, dehydrated, and embedded in paraffin wax. Tissue sections (3 µm) were stained with hematoxylin and eosin (H&E). The changes in muscle morphology were visualized using a light microscope (Olympus CX21, Hicksville, NY, USA) and the ZEN software (Axis Cam ERc5s-Zen lite, 2012). The ImageJ program was used for muscle fiber diameter analysis modifying the procedure described in a previous study [29].

#### 2.3.7. Proximate Analysis

Proximate analysis of meat was performed following the standardized method of the Association of Official Analytical Chemists [30]. Briefly, 2 g breast and thigh meat samples were both dried at 102 °C for 15 h to estimate the moisture content. The crude protein (CP) percentage was determined using the Kjeldahl method (VAPO45, Gerhardt Ltd., Idar-Oberstein, Germany), and the total crude fat content was determined following the protocol of Jeon et al. [31].

#### 2.3.8. Fatty Acid Profile Measurement

Total lipids were extracted from breast and thigh samples. Briefly, 5 g breast and thigh meat samples were both dissolved in 90 mL chloroform–methanol (2:1, *v*/*v*), and total lipid was extracted following the method described in a previous study [32]. Subsequently, fatty acid methyl esters (FAMEs) were prepared by methylation following the procedure reported by Metcalfe et al. [33]. The FAMEs were analyzed using gas chromatography (Hewlett-Packard 7890A; Agilent Technologies, Santa Clara, CA, USA) fitted with a capillary column (SP 2560, Supelco Inc., Bellefonte, PA, USA, 100 m × 0.25 mm i.d., 0.20-µm film thickness) and a flame ionization detector. Helium was used as the carrier gas at a flow rate of 0.95 mL/min. The temperatures of the injector and detector were set at 260 °C. The injector and detector temperatures were set to 260 °C. The oven temperature was programmed to increase from 70 °C to 175 °C at a rate of 13 °C/min, and then rise to 240 °C at a rate of 4 °C/min.

#### 2.3.9. Total Collagen Content Measurement

Total collagen content was estimated following the method described by da Silva et al. [34] with some modifications. Briefly, 50 mg breast and thigh meat samples were both hydrolyzed in 1 mL 7 M NaOH in an autoclave at 121 °C for 40 min. Sulfuric acid (3.5 M) was used to neutralize the hydrolyzed samples to a pH of 7. Then, the neutralized samples were filtered and mixed with chloramine T solution and Ehrlich’s reagent. Afterward, the absorbance at 550 nm (Genesys 10S UV-VIS, Thermo Fisher Scientific, Madison, WI, USA) was measured using hydroxyproline (Sigma-Aldrich Co., St. Louis, MO, USA) as a standard. A coefficient of 7.25 was used to calculate the total collagen content [35]. The collagen content was expressed in mg of collagen per g of meat.

#### 2.3.10. Nucleotide Content Measurement

To extract the nucleic acids, 5 g breast and thigh meat samples were both mixed with 30 mL ice-cold 7.5% perchloric acid and homogenized for 30 s. Next, 10 mL ice-cold 7.5% perchloric acid was added and centrifuged at 2000× *g* at 4 °C for 5 min. The solution was then filtered through a filter paper (No.1, Whatman International Ltd., Maidstone, UK). The filtrate (1 mL) was analyzed using high-performance liquid chromatography (HPLC) (HP 1260, Agilent Technologies, Inc., Santa Clara, CA) fitted with a Hypersil ODS C18 column (3 µm, 150 mm × 4.6 mm) (Thermo Scientific, Waltham, MA, USA). The analytical conditions for HPLC were set following Kim et al. [36] with some modifications. The peaks of the individual nucleotides were identified using the retention times estimated for the standards: inosine-5′-monophosphate (IMP) and guanosine-5′-monophosphate (GMP) (both obtained from Sigma, St. Louis, MO, USA), and the concentration of each nucleotide was estimated from the peak area of the individual nucleotides.

#### 2.3.11. Cholesterol Measurement

The cholesterol content in the meat samples was estimated by gas chromatography following the method described by Rowea et al. [37] with some modifications. The α-cholesterol was used as the internal standard. The gas chromatograph fitted with a flame ionization detector equipped with an HP-5 column (30 m × 0.32 mm; film thickness, 0.22 µm; Agilent Technologies, Palo Alto, CA, USA) was used for the analysis. The injection port and detector temperature were set at 260 °C and 255 °C, respectively. Cholesterol was identified by comparing the relative retention time of the sample with that of the standard (Cargo Erba Reagents, Milan, Italy).

### 2.4. Synchrotron Radiation-Based Fourier Transform Infrared (SR-FTIR) Spectroscopy

#### 2.4.1. Sample Preparation

Breast samples were cut into 1 cm × 1 cm pieces and placed in an aluminum foil block filled with optimal cutting temperature (OCT). Subsequently, the cut samples were completely embedded in OCT and immediately fixed in liquid nitrogen. The breast samples were then cut into sections using a cryostat (micron/HM 525) until the region of interest was reached. The optimized thicknesses of the tissue sections were 6 μm for infrared measurement. The breast sample sections were then kept in a desiccator with a vacuum pump for 30 min.

#### 2.4.2. SR-FTIR Spectra Measurement

The biochemical composition of the samples was analyzed using SR-FTIR spectroscopy [38]. Spectral data were collected using the infrared microspectroscopy beamline BL4.1 IR Spectroscopy and Imaging at the Synchrotron Light Research Institute (SLRI, Nakhon Ratchasima, Thailand). Spectra were obtained using a Vertex 70 FTIR spectrometer (Bruker Optics, Ettlingen, Germany) coupled to an IR microscope (Hypersion 2000, Bruker), equipped with a liquid nitrogen cooled MCT detector. The data were collected over the 4000 to 800 cm^−1^ measurement range. The measurement was performed in mapping mode with an aperture size of 10 µm × 10 µm and acquisition of 64 scans with a spectral resolution of 4 cm^−1^. The software OPUS 7.2 (Bruker Optics Ltd, Ettilngen, Germany) was used for the derivation of the spectra and instrument control, and the results were analyzed with the CytoSpec software.

Samples from CO and OR (12 samples per group) were used to investigate the changes in the biochemical composition of meat. First, the original spectra were averaged to obtain a total of five spectra, followed by the second derivation at 13 smoothing points, and the vector was normalized using the Savitzky–Golay method in Unscrambler X software (version 10.1, Camo Analytics, Oslo, Norway) to account for the effects of varying sample thickness.

#### 2.4.3. Relative Integral Area for Each Functional Group

The relative integral areas were calculated from the second derivative spectra in the spectral region from 3000 to 900 cm^−1^ as follows: 3000 to 2800 cm^−1^ (CH stretching of lipid), 1740 cm^−1^ (C=O ester of lipid), 1700 to 1600 cm^−1^ (amide I), 1600 to 1500 cm^−1^ (amide II), 1338 cm^−1^ (amide III), and 1250 to 900 cm^−1^ (carbohydrate and glycogen) using OPUS software (version 7.2, Bruker Optics Ltd.), as shown in Table 2.

#### 2.4.4. Curve Fitting for the Amide I Band

The peak positions and band shapes were selected for curve fitting, which examines the area of overlapping peaks of the amide I band (1700 to 1600 cm^−1^) in the FTIR spectra using a nonlinear least square approach based on Gaussian and Lorentzian functions. The fitting parameters, such as beta-sheet (1645 to 1620 cm^−1^), alpha-helix (1640 to 1650 cm^−1^), beta-turn (1685 to 1675 cm^−1^), and anti-parallel (1695 to 1685 cm^−1^) were measured.

### 2.5. Statistical Analyses

The significant differences of the mean values of growth performance traits, carcass composition, breast meat quality, biochemical compositions, and FA content of KRC meat between CO and OR were analyzed by t-test using SPSS software (version 16.0; SPSS Inc., Chicago, IL, USA). All data are expressed as mean ± SD, and a *p*-value of < 0.05 was considered significant. 

The interaction of the spectral data matrix between CO and OR chicken samples, meat quality, n-3 PUFA content, secondary protein structures, and biochemical compounds (spectra intensity from 3000 to 1000 cm^−1^) from FTIR was generated. The clustering of the variables was analyzed using principal component analysis (PCA). The relationships between variables and sample properties were identified using biplot obtained by the calculations from a two-dimensional scatter plot of PCA with the dominant spectral band of the different variables.

The correlation between meat quality, n-3 PUFA content, secondary protein structures, and biochemical compounds for each cluster of the control and organic chicken samples in the data matrix were weighted using an SD weighting process and calculated using PCA, after which a biplot correlation between variables was created using multivariate analysis.

## 3. Results and Discussion

### 3.1. Growth Performance and Carcass Yield

The growth performance and carcass yield of chickens are shown in Table 3 and Table 4, respectively. It was observed that the final BW of the OR and CO chickens did not differ (*p* > 0.05), whereas the FI and FCR of the OR chickens were lower than those of CO (*p* < 0.001). The results were inconsistent with our hypothesis that chickens reared in OR would be exposed to fluctuating temperature and increased activity in the yard requiring higher energy, consequently leading to decreased BW and increased FCR. Our hypothesis aligns with the findings of Mattioli et al. [46], who reported that exercise behavior is negatively correlated with the performance of chickens, as high movement can increase energy metabolism and decrease their growth performance. Although several studies related to the effect of OR on growth performance have been reported [12,47,48], the results of these studies, including those of the present study, show inconsistency, which could be attributed to the differences in the rearing environmental factors, including light intensity, photoperiod, temperature, breed of chicken, diet, forages, insects, and worms found in pasture [20]. Likewise, several studies have shown that the raising environment affects the quality of grass [49], natural diet [50,51], and diversity of microorganisms in a specific area [52]. Furthermore, in concordance with the results of previous studies, the reduced FI of OR chickens could also be attributed to their free access to natural diets from the rearing environment [10,53]. The lower FCR in OR chickens than in CO chickens with no significant differences in their BW could be due to the enrichment of the digestive tract of OR chicken with some beneficial microorganisms, which might have contributed to the activation of the beneficial enzymes, consequently leading to increased utilization of protein or carbohydrate from grass [54]. However, as the present study was aimed to identify the effects of raising systems on growth performance, we did not explore the precise role of different feed sources such as natural or commercial diets on gut microbiota composition. Therefore, further in-depth studies are needed to understand the effect of OR on the gut microbiota of chicken.

Furthermore, the findings demonstrated no differences (*p* > 0.05) in carcass yields of the OR and CO chickens, whereas the yield of abdominal fat in chickens from OR was lower (*p* = 0.029) than that of the chickens from CO, as reported in previous studies [12,55]. Though it was expected that the carcass yield of chickens in OR would be higher than that of chickens in CO because the formers have more activity during the day, resulting in the process of muscle repair and increased muscle fiber size (hypertrophy), our study, like the study of Castellini et al. [12], did not adhere to the above expectations. This could be due to the high temperature (20.0–35.5 °C) during the experimental period (summer) of this study that might have restricted the chickens to stay close to their house, leading to reduced exercise and motor activities; hence, no gain in muscle mass. In contrast, Comert et al. [9] have demonstrated a higher amount of abdominal fat in chicken grown in the OR system than those grown in the CO system. This difference could be attributed to the different genotypes and the sex of birds used in the two studies. Taken together, we inferred that the OR chickens are exposed to increased physical activity than the CO chickens, which, though increases the energy metabolism rate and reduces the abdominal fat accumulation, is not sufficient enough to increase the carcass yield. Furthermore, sex and genotype are the other important factors affecting the carcass characteristics and, therefore, should be carefully considered [9,56].

### 3.2. Physicochemical Properties of Chicken Meat

The effects of the raising system on the biochemical composition of OR chicken meat are presented in Table 5. The results demonstrated that except for protein and total collagen content, the raising system had no effects (*p* > 0.05) on moisture, cholesterol, fat, IMP, and GMP contents in breast and thigh meat. However, the protein and total collagen contents were higher (*p* < 0.05) in OR meat than in CO meat.

IMP and GMP are key compounds contributing to flavor [57]; in addition, they participate in energy metabolism and ensure energy supply to cells. IMP is generated from the process of adenosine triphosphate (ATP) consumption [58], and ATP is produced when an animal has any activity [58,59]. Moreover, IMP can be converted to GMP [60]. Considering these, it can be inferred that the physical activities might not be sufficiently different in OR chicken than the CO chicken; therefore, no significant differences were observed for IMP, GMP, cholesterol, and crude fat between OR and CO chicken meat samples.

On the contrary, the increased protein and collagen contents suggested that the access to the outdoors increased the physical activity in OR chickens, which was sufficient to alter these components, and indicated that these two traits could be highly sensitive to physical activities. In concordance with the above speculation, Miller et al. [61] have reported that the rate of skeletal muscle collagen and sarcoplasmic protein synthesis increased markedly and rapidly after exercise. Moreover, the results of the present study are congruent with those of the previous studies [5,6,62]. Mikulski et al. [62] reported that the highest protein content was detected in the meat of chicken raised in outdoor access, about 12 h a day, from d 21 to d 64. It has been shown that organic chickens which forage on pasture 12 h daily, (depending on the condition each day) exhibited greater activity from d 21 to d 84, resulting in more type IIA muscle fibers, and were able to synthesize more protein [63]. Collectively, the findings of the present study and those of the previous studies suggest that rearing chickens with outdoor access could increase CP and total collagen content in meat.

The results for the physicochemical properties of meat shown in Table 6 indicate that raising systems had no significant effect on most traits, except for shear force and color, whose values were higher (*p* < 0.001) in OR chicken than in CO chicken.

Generally, the ultimate pH is largely determined by the initial glycogen storage in the muscle, and the decline in muscle pH is related to glycolysis activity under anaerobic conditions [64], wherein lower pH is related to higher drip loss and cooking loss [65]. This could be because a decline in the muscle pH causes a reduction in the net charge of muscle protein and charged protein sites for binding of water molecules, resulting in greater water and nutrient losses [66]. However, in this study, the raising systems demonstrated no effect on pH, drip loss, and cooking loss. These results suggested that the energy expenditure might not significantly affect the rate of glycolysis in the chickens from both raising systems.

Shear force and muscle diameter indicate tenderness [67]. Increased shear force is the consequence of higher protein and collagen levels in the meat sample. Furthermore, it is known that chicken breast meat is composed of type IIB muscle fibers [59], and prolonged exercise training can induce the transition of type IIB muscle fibers to type IIA muscle fibers. The latter fiber type has a high capacity to generate ATP by oxidative metabolic processes. Therefore, it requires more oxygen to maintain its activity and induce protein synthesis, leading to increased muscle diameter [68,69]. In this study, the muscle diameter of OR chicken meat was slightly larger (*p* = 0.056) than CO chicken, which indicated that the OR chickens have higher movement, though not significant, leading to adaptive changes in the skeletal muscle fiber.

The greater redness (*p* = 0.004) and yellowness (*p* < 0.0001) of OR meat and skin observed in this study agree with those reported in the study of Grashorn and Serini [5]. It could be attributed to the consumption of grasses, a major source of carotenoid pigments [16,62,70], as the OR chickens had free access to the grass fields.

### 3.3. Fatty Acid Profile of Chicken Meat

The FA profiles in the breast and thigh meat of OR and CO chickens are shown in Table 7. No differences in saturated fatty acids (SFA), monounsaturated fatty acids (MUFA), and PUFA content of KRC meat were detected in the different raising systems (*p* > 0.05). In contrast, the proportion of total n-3 PUFA in breast and thigh meat of chickens raised under OR was higher (*p* = 0.01) than in those raised under CO. Moreover, the DHA (C22:6n-3) content was higher (*p* = 0.01) in breast meat. However, the ratio of n-6 to n-3 PUFA was lower in the breast (*p* < 0.001) and thigh (*p* = 0.02) of OR chickens. It has been shown that FA from the diet strongly influences the amount of FA in meat [71]. Similarly, grass intake has been shown to increase antioxidants content in plasma and consequently decrease FA oxidation [45]. In addition, fresh grass contains 50–75% ALA [72], a precursor of long-chain n-3 PUFA. It can be converted to EPA (C20:5n-3), docosapentaenoic acid (DPA, C22:5n-3), and DHA (C22:6n-3) through the biochemical processes of elongation and desaturation [14]. Moreover, previous studies reported that slow-growing chickens have higher expression of FADS1 and FADS2 genes involved in n-3 PUFA and n-6 PUFA metabolism [73] and can maintain their oxidative stability during their activity than fast-growing chickens [46]. Considering these, the increased level of DHA in KRC breast meat could be attributed to the influence of the consumption of pastures by the OR chickens. At the same time, the non-increase in DHA content in the thigh meat could be explained by the different lipid compositions in breast and thigh meat. The DHA content is preferentially stored in the form of phospholipids than triglycerides. These results are consistent with Bou et al. [74], who reported that the ratio of phospholipids to triglycerides is higher in breast meat than in thigh meat.

Furthermore, a lower ratio of n-6 to n-3 PUFA in the meat of OR chickens owing to increased n-3 PUFA levels could also be due to the competition of FA for desaturase and elongase enzymes [75]. In concordance, Lopez-Ferrer et al. [76] have also reported that the high levels of n-3 PUFA intake may have reduced the desaturase and elongase enzymes of the precursors of n-6 PUFA, leading to low n-6 to n-3 PUFA in organic KRC meat. Interestingly, OR chicken showed a ratio of n-6 to n-3 lower than 10, which is beneficial for human health [77].

Dal Bosco et al. [10] reviewed several studies and reported an interaction between the genetic strain with movement, intake of antioxidants, the antioxidant capacity of the body, plasma, and fatty acid profile meat. The study demonstrated that selected strains, higher kinetic activity, and the less controlled environmental conditions exacerbate oxidative status and fatty acid profile of the meat. Therefore, this relationship must be analyzed in future studies. Measurement of enzyme activity and expression of genes involved in FA metabolisms, such as FADS1 and FADS2, and antioxidant status, to obtain a solid and sound knowledge to explain FA accumulation in organic KRC as a representative of slow-growing chickens on OR are needed.

### 3.4. Changes in Biochemical Profile and Secondary Protein Structure in Breast Meat

The average original and second derived SR-FTIR spectra in the fingerprint region of wavenumbers from 3000 to 900 cm^−1^ obtained from CO and OR are shown in Figure 1A,B, respectively. The difference in the average spectra of breast meat samples from CO and OR detected in the ranges of 2946 cm^−1^ and 2885 cm^−1^ represent lipid; 1700 cm^−1^ and 1672 cm^−1^ represent protein (amide I) and 1139 to 955 cm^−1^ represent carbohydrate and glycogen, respectively.

The areas under the peaks in these regions were integrated and calculated, revealing differences in the CO and OR spectra (Table 8). The integral areas of lipid (C-H stretching), amide I (C=O stretching), amide II (C-N stretching vibrations in combination with N-H bending), and glycogen regions were greater (*p* < 0.05), whereas no significant difference (*p* = 0.061) was found in amide III in OR compared to those in CO. On the contrary, the result of amide III obtained from the t-test did not agree with the result of the loading correlation, as shown in Figure 2B. These differences are due to the fact that PCA consists of the analysis of data sets containing imprecise measurements, which only extracts the maximum variance and important information from the data set. It presents the result as a set of summary indicators known as principal components showing the differences in the data matrix [26]. The amide I band represents different types of secondary protein structures such as α-helix, β-sheet, β-turn, and antiparallel β-sheet, which are related to meat quality [78]. Here, we examined the secondary structures of proteins in the region of the amide I band, where we found a significant difference and calculated the ratio of their relative contents by curve fitting (Table 9). The results revealed no differences (*p* > 0.05) in secondary protein structure between the OR and CO chickens, except that the β-sheet of OR chicken meat was slightly lower (*p* = 0.066) than that of CO. A possible reason for this could be that the different fat and FA profile between the two groups could have altered the rate of lipid oxidation leading to different conformational changes in the secondary protein structures [18].

### 3.5. Correlation Loading Plot of FTIR Spectra with the Biochemical Compounds and Quality of Breast Meat from Different Raising Systems

To elucidate the relationship between biochemical compounds and the quality of breast meat (Figure 2A) from the two raising systems, we combined data from FTIR spectra related to biomolecules and physicochemical properties of chicken meat and then used PCA to classify the CO and OR groups. As shown in Figure 2A, the meat of chickens from different raising systems was separated, representing 57% of the total variability of all data sets.

The correlation loading plot in Figure 2B shows traits, shear force, meat color (redness; a*, yellowness; b*), skin color (lightness; L*^s^), CP, lipid, collagen, carbohydrate and glycogen, amide I, amide II of protein, and amide III of collagen located in the outer circle areas, which explained over 50% of the variance between the two groups that had significant correlations among these traits.

The negative loading plot in Figure 2B shows the shear force, meat color (a* and b*), CP, collagen content, amide I of protein (1700 to 1600 cm^−1^), amide II of protein (1480 to 1575 cm^−1^), amide III of collagen (1229 to 1310 cm^−1^), carbohydrate and glycogen (1200 to 1000 cm^−1^), and lipid (2955 to 2800 cm^−1^) were positively correlated with negative score plot from chicken breast meat in the OR group. Although the amide I of β-turn (1670 to 1680 cm^−1^) was in the outer circle, its position was close to the PC-2 axis of the loading plot, meaning that they could not be used to distinguish breast meat from CO and OR. On the contrary, the positive loading plot showed that skin color (lightness; L*^s^) was positively correlated with the positive score plot of chicken breast meat from CO. The results revealed that the raising system could differentiate meat properties.

The PCA results confirmed the results shown in Table 4 and Table 5. In addition, glycogen and lipids identified by SR-FTIR were higher in the breast meat of OR chicken than those in CO chicken. This could be explained by the fact that organic chickens have high activity and more glycogen storage in their skeletal muscle [79].

Furthermore, under excess glycogen stores, glucose can be converted to fat, which is stored in the muscle by de novo lipid synthesis [80]. Therefore, we analyzed the relationship between raising systems and FA composition in chicken meat using PCA. As shown in Figure 2C, the FA profiles in chicken breast and thigh meat from different raising systems were separated with 50% of the total variability of all data sets. The correlation loading plot in Figure 2D shows the traits, total SFA, PUFA, linoleic acid (LA; C18:2n-6), arachidonic acid (C20:4n-6; AA), and n-3 PUFAs such as ALA and DHA located in the outer circle areas, which explained over 50% of the variance between the two groups that had significant correlations among these traits.

The negative loading plot in Figure 2D shows that PUFA, n-3 PUFA in the breast and thigh, ALA, SFA, and LA in thigh and DHA in breast meat were positively correlated with the negative score plot from chicken meat in the OR group. The positive loading plot in Figure 2D shows that the ratio of n-6 to n-3 PUFA was positively correlated with the positive loading plot of CO but negatively correlated with the negative score plot of chicken meat from OR.

The PCA results confirmed some FA components in chicken meat from the OR, as shown in Table 7. Moreover, the PCA result explored the correlation between OR and amount of SFA in thigh meat, which could be attributed to the FA composition of the feed, especially Ruzi grass that contains 22.91% SFA. In addition, the analysis revealed higher contents of PUFA and ALA and a lower content of n-6 PUFA in OR. These results further supported the conclusion that feed plays an important role in modifying the FA profile of chicken meat. Furthermore, it has been shown that high amounts of PUFA and ALA in Ruzi grass can modify PUFA and ALA in meat, while an increased intake of PUFA, especially FA from the n-3 PUFA group, can reduce n-6 PUFA such as LA in breast and AA in thigh meat [81].

## 4. Conclusions

In conclusion, our results reveal that OR has no negative effect on the growth performance of slow-growing chickens. The study shows that the OR system has a positive effect on the meat characteristics, especially meat color and texture, biochemical compounds such as proteins (amide I and amide II), total collagen (amide III), and beneficial FA (PUFA, DHA, and ALA), which determine the nutritional value of meat. The findings of this study demonstrate the potential of OR for commercial adoption in tropical regions such as Southeast Asia. Furthermore, the study demonstrates the efficiency of SR-FTIR to determine the differences in the biochemical compounds, which could serve as markers to monitor meat quality traits. Collectively, these findings provide insights into the relative roles of raising systems in KRC chickens and can help producers to produce nutritionally rich quality products while maintaining animal welfare standards.

## Figures and Tables

**Figure 1 animals-12-00570-f001:**
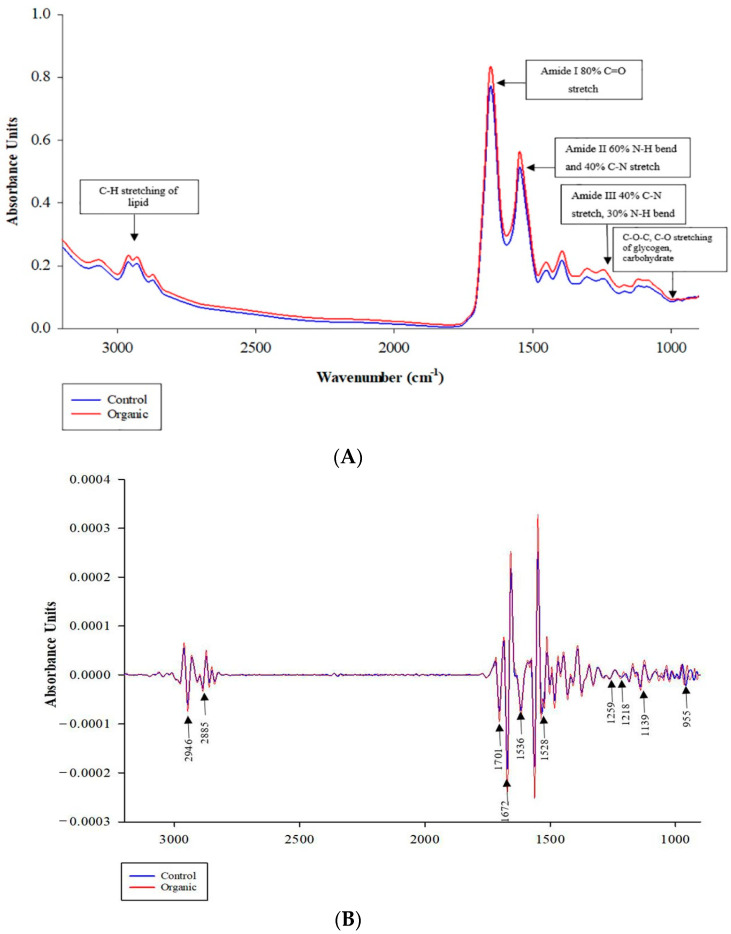
Average SR-IR spectra of original (**A**) and second derivative (**B**) from KRC chicken breast meat between control and organic raising systems. Infrared spectra were collected in the spectral range of 4000 to 900 cm^−1^, resolution 4 cm^−1^ based on 300 spectra per treatment. 3000 to 2800 cm^−1^ (CH stretch of lipid); 1740 cm^−1^ (C=O ester of lipid); 1700 to 1600 cm^−1^ (amide I); 1600 to 1500 cm^−1^ (amide II); 1338 cm^−1^ (amide III); and 1250 to 900 cm^−1^ (carbohydrate and glycogen) parameters used for the collected spectra.

**Figure 2 animals-12-00570-f002:**
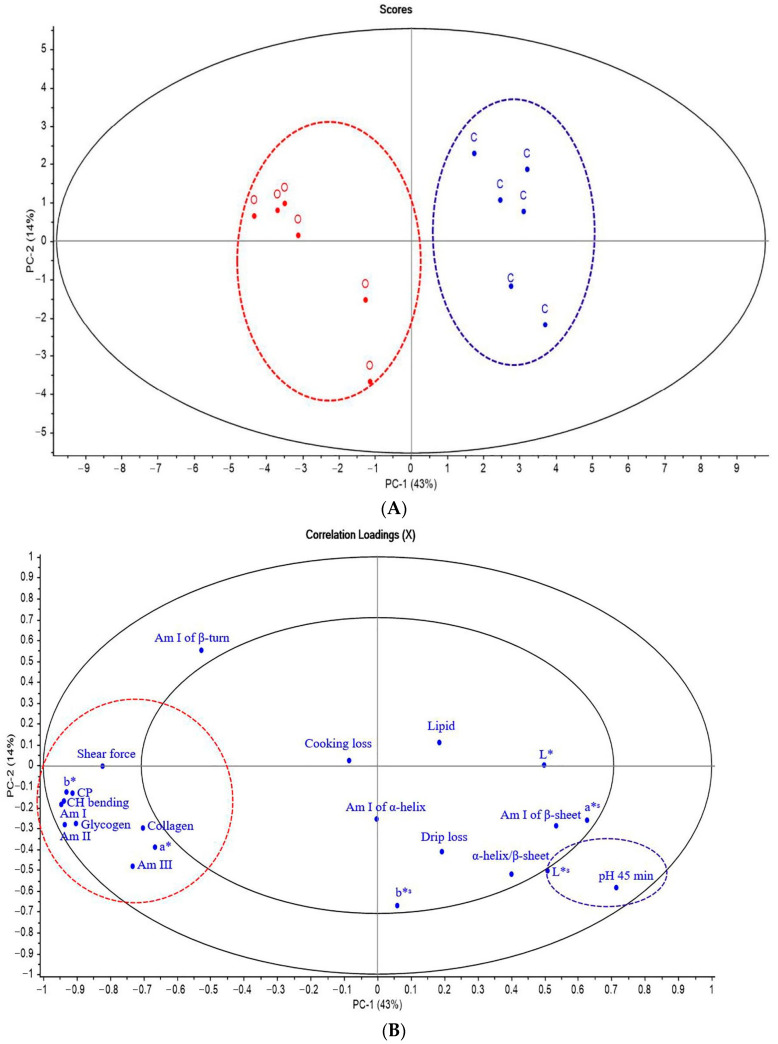
Principal component analysis (PCA) score plot and correlation loading plot (PC-1 vs. PC-2) of SR-FTIR spectra, biochemical compounds and meat quality (**A**,**B**) and fatty acid in breast and thigh meat (**C**,**D**) of KRC in different production systems. Abbreviations: C, control group; O, organic group; Am I, amide I of protein; Am II, amide II of protein, Am III, amide III of collagen, CP, crude protein; a*, redness; b*, yellowness; L*, lightness; a*^s^, redness of skin; b*^s^, yellowness of skin; L*^s^, lightness of skin; SFA, saturated fatty acids; PUFA, polyunsaturated fatty acids; MUFA, monounsaturated fatty acids; α-linolenic acid; DHA, docosahexaenoic acid; EPA, eicosapentaenoic acid; LA, linoleic acid; # = thigh meat, without # = breast meat.

**Table 1 animals-12-00570-t001:** Compositions and calculated nutrient contents of starter, grower, and finisher diets (g/100 g diet, as-fed basis).

Item	Starter (d 1 to d 21)	Grower (d 22 to d 42)	Finisher (d 43 to d 84)
Full fat soybean meal (37% CP)	47.70	41.00	34.50
Broken rice	48.50	55.65	62.35
DL-methionine	0.25	0.10	0.10
Salt	0.35	0.35	0.35
CaCO_3_	1.40	1.30	1.20
Monocalcium phosphate (21% P)	1.30	1.10	1.00
Premix ^1^	0.50	0.50	0.50
Calculated nutrients (% unless stated otherwise)			
ME (kcal/kg)	3175	3190	3195
Crude protein	21.00	19.00	17.00
Crude fat	8.33	7.00	7.18
Crude fiber	2.73	2.37	2.05
Digestible lysine	1.21	1.08	0.95
Digestible methionine	0.59	0.43	0.41
Digestible met + cys	0.93	0.73	0.69
Digestible threonine	0.79	0.72	0.64
Calcium	1.01	0.91	0.84
Available phosphorus	0.45	0.38	0.35

^1^ Premix (0.5%) provided the following per kilogram of diet: 15,000 IU of vitamin A; 3000 IU of vitamin D3; 25 IU of vitamin E; 5 mg of vitamin K3; 2 mg of thiamine; 7 mg of riboflavin; 4 mg of pyridoxine; 25 µg of cobalamin; 11.04 mg of pantothenic acid; 35 mg of nicotinic acid; 1 mg of folic acid; 15 µg of biotin; 250 mg of choline chloride; 1.6 mg of copper; 60 mg of manganese; 45 mg of zinc; 80 mg of iron; 0.4 mg of iodine; 0.15 mg of selenium.

**Table 2 animals-12-00570-t002:** Relationship between molecular functional group and biomolecules.

Wavenumber (cm^−1^)	Chemical Function	Assignment	References
2930 to 2910	CH_2_ asymmetric stretch	Mainly saturated lipids, proteins	[39]
2970, 2957 to 2953	CH_3_ asymmetric stretching	Lipids (mainly), proteins	[40]
2875 to 2870	CH_3_ symmetric stretching	Lipids, proteins	
1700 to 1600	C=O stretching	Amide I band of proteins	[41]
1655, 1650 to 1640	C=O stretching	Amide I of α-helical structures of proteins	[42,43]
1695 to 1685	C=O stretching	Antiparallel β-sheet	[41]
1550 to 1520	C-N stretching + N-H bending coupled in of face	Amide II band of proteins	[43]
1310 to 1240	C-N stretching + N-H bending coupled in of face	Amide III of proteins	[40,43]
1637 to 1615	C=O stretching	β-sheet	[40,43,44]
1681 to 1664 1685 to 1675	C=O stretching	β-turn	[44,45]
1200 to 900	C-O-C, C-O dominated by ring vibrations of carbohydrates C-O-P, P-O-P	Carbohydrate and Glycogen	[40]

**Table 3 animals-12-00570-t003:** Effects of organic raising system on growth performance of Korat chicken at 84 d of age (mean ± SD).

Item	CO	OR	*p*-Value
Final BW (g)	1480.31 ± 37.94	1445.56 ± 42.90	0.558
FI (g)	3634.74 ± 32.22 ^a^	3074.69 ± 21.37 ^b^	<0.0001
BWG (g)	1434.60 ± 37.74	1399.68 ± 42.80	0.554
FCR	2.54 ± 0.06 ^a^	2.20 ± 0.07 ^b^	0.004

^a,b^ Means within a row with different superscript letters differ significantly at *p <* 0.05. CO = control, OR = organic raising system.

**Table 4 animals-12-00570-t004:** Effects of organic raising system on carcass composition of Korat chicken at 84 d of age (mean ± SD).

Yield (%)	CO	OR	*p*-Value
Eviscerated carcass ^1^	63.90 ± 0.94	63.67 ± 0.84	0.941
Pectoralis minor	3.27 ± 0.12	3.05 ± 0.13	0.247
Pectoralis major	8.64 ± 0.20	8.49 ± 0.30	0.680
Thigh meat	10.06 ± 0.26	10.38 ±0.35	0.464
Drumstick meat	10.02 ± 0.37	9.68 ± 0.30	0.477
Abdominal fat	1.47 ± 0.15 ^a^	1.05 ± 0.11 ^b^	0.029

^a,b^ Means within a row with different superscript letters differ significantly at *p <* 0.05. ^1^ Carcass weights without viscera, head, neck, feet, and shank. CO = control, OR = organic raising system.

**Table 5 animals-12-00570-t005:** Effects of organic raising system on biochemical composition of Korat chicken at 84 d of age (mean ± SD).

Item	Treatment		*p*-Value
	CO	OR	
**Breast meat**			
Moisture (%)	73.74 ± 1.20	72.89 ± 0.92	0.580
Crude Protein (%)	23.58 ± 0.12 ^b^	24.54 ± 0.07 ^a^	<0.0001
Crude fat (%)	1.99 ± 0.10	1.93 ± 0.15	0.430
Cholesterol (mg/100 g meat)	59.04 ± 6.91	52.68 ± 4.78	0.450
Total collagen (mg/g meat)	0.85 ± 0.05 ^b^	1.02 ± 0.05 ^a^	0.030
IMP (mg/g meat)	0.14 ± 0.00	0.13 ± 0.00	0.150
GMP (mg/g meat)	4.93 ± 0.12	5.09 ± 0.09	0.150
**Thigh meat**			
Moisture (%)	74.04 ± 0.30	73.03 ± 0.14	0.060
Crude Protein (%)	13.74 ± 0.04 ^b^	14.44 ± 0.04 ^a^	<0.0001
Crude fat (%)	5.64 ± 0.35	5.43 ± 0.35	0.508
Cholesterol (mg/100 g meat)	79.89 ± 9.01	76.37 ± 5.74	0.720
Total collagen (mg/g meat)	0.73 ± 0.03 ^b^	1.08 ± 0.06 ^a^	<0.0001
IMP (mg/g meat)	0.15 ± 0.00	0.16 ± 0.00	0.220
GMP (mg/g meat)	3.73 ± 0.22	4.18 ± 0.28	0.100

^a,b^ Means within a row with different superscript letters differ significantly at *p* < 0.05. CO = control, OR = organic raising system.

**Table 6 animals-12-00570-t006:** Effects of organic raising system on breast meat quality of Korat chicken at 84 d of age (mean ± SD).

Item	CO	OR	*p*-Value
Ultimate pH	5.40 ± 0.02	5.35 ± 0.02	0.079
Drip loss (%)	11.93 ± 0.66	12.27 ± 0.62	0.874
Cooking loss (%)	22.84 ± 0.95	23.09 ± 0.55	0.570
Shear force (WBS)	2.17 ± 0.04 ^b^	2.63 ± 0.10 ^a^	<0.0001
Skin color			
Lightness	66.96 ± 0.59	65.34 ± 2.47	0.057
Redness	−0.47 ± 0.19 ^b^	0.30 ± 0.15 ^a^	0.004
Yellowness	7.02 ± 0.37 ^b^	15.50 ± 0.66 ^a^	<0.0001
Meat color			
Lightness	52.21 ± 0.65	51.37 ± 0.55	0.591
Redness	−0.29 ± 0.18 ^b^	0.35 ± 0.13 ^a^	0.031
Yellowness	3.49 ± 0.28 ^b^	7.27 ± 0.44 ^a^	<0.0001
Muscle diameter (µm)	22.34 ± 0.51	23.88 ± 0.56	0.056

^a,b^ Means within a row with different superscript letters differ significantly at *p* < 0.05. CO = control, OR = organic raising system. WBS = Warner-Bratzler shear force (kgf/0.5 cm^2^).

**Table 7 animals-12-00570-t007:** Major fatty acid profiles (g/100 g total FA) of skinless breast and thigh meat from organic chickens (mean ± SD).

Fatty Acid	Breast Meat			Thigh Meat		
	CO	OR	*p*-Value	CO	OR	*p*-Value
C14:0	0.90 ± 0.37	0.96 ± 0.48	0.92	0.39 ± 0.01	0.38 ± 0.02	0.79
C16:0	18.27 ± 0.22	17.54 ± 0.30	0.06	16.81 ± 0.46	16.99 ± 0.48	0.79
C16:1	1.53 ± 0.16	1.16 ± 0.25	0.23	2.01 ± 0.24	2.09 ± 0.25	0.82
C18:0	8.28 ± 0.13	8.63 ± 0.26	0.24	7.61 ± 0.36	7.42 ± 0.21	0.66
C18:1n-9	24.56 ± 0.89	25.78 ± 1.50	0.65	26.66 ± 0.75	28.67 ± 0.83	0.07
C18:2n-6	31.09 ± 0.20	28.91 ± 0.11	0.11	37.45 ± 1.11	34.90 ± 0.99	0.49
C18:3n-6	0.13 ± 0.08	0.11 ± 0.09	0.87	0.26 ± 0.01	0.22 ± 0.02	0.30
C18:3n-3	2.12 ± 0.15	2.25 ± 0.21	0.62	3.51 ± 0.16	3.77 ± 0.14	0.24
C20:2n-6	0.30 ± 0.03	0.35 ± 0.04	0.30	0.28 ±0.02	0.28 ± 0.04	0.90
C20:3n-6	0.53 ± 0.04	0.63 ± 0.05	0.13	0.24 ± 0.02	0.27 ± 0.01	0.33
C20:4n-6	9.58 ± 0.86	10.50 ± 0.91	0.21	3.98 ± 0.44	3.15 ± 0.45	0.20
C20:5n-3	0.13 ± 0.05	0.16 ± 0.09	0.72	0.16 ± 0.11	0.03 ± 0.02	0.29
C22:6n-3	1.11 ± 0.07 ^b^	1.75 ± 0.19 ^a^	0.01	0.41 ± 0.05	0.67 ± 0.13	0.09
SFA	27.73 ± 0.50	27.13 ± 0.58	0.32	24.80 ± 0.59	24.79 ± 0.52	0.40
MUFA	27.29 ± 1.00	28.82 ± 1.62	0.29	28.89 ± 0.89	31.08 ± 0.96	0.07
PUFA	44.98 ± 0.83	44.04 ± 0.85	0.31	46.31 ±1.14	44.13 ± 1.36	0.27
Total n-6	40.79 ± 0.77	39.87 ± 0.83	0.18	41.94 ± 1.02	38.59 ± 0.94	0.19
Total n-3	3.36 ± 0.14 ^b^	4.17 ± 0.18 ^a^	< 0.001	4.09 ± 0.15 ^b^	5.27 ± 0.18 ^a^	0.01
n-6/n-3	12.14 ± 0.54 ^a^	9.57 ± 0.40 ^b^	< 0.001	10.26 ± 0.21 ^a^	7.33 ± 0.63 ^b^	0.02

^a,b^ Means within a row with different superscript letters differ significantly at *p* < 0.05. CO = control, OR = organic raising system, SFA = saturated fatty acids, MUFA = monounsaturated fatty acids, PUFA =polyunsaturated fatty acids.

**Table 8 animals-12-00570-t008:** The integral area of average spectra of KRC breast meat from different raising systems (mean ± SD).

Biomolecule (Wavenumber)	CO	OR	*p*-Value
C-H stretching of lipid	1.13 ± 0.005 ^a^	1.10 ± 0.005 ^b^	0.017
Amide I 80% C=O stretch	11.38 ± 0.007 ^b^	11.97 ± 0.006 ^a^	<0.0001
Amide II 60% N-H bend and 40% C-N stretch	0.33 ± 0.001 ^b^	0.58 ± 0.001 ^a^	<0.0001
Amide III 40% C-N stretch, 30% N-H bend	0.74 ± 0.004	0.78 ± 0.006	0.061
C-O-C, C-O stretching of glycogen, carbohydrate	0.29 ± 0.002 ^b^	0.38 ± 0.002 ^a^	<0.0001

^a,b^ Means within a row with different superscript letters differ significantly at *p* < 0.05. CO = control, OR = organic raising system.

**Table 9 animals-12-00570-t009:** Ratio of relative content of secondary structures determined by SR-IR from KRC organic chickens breast meat.

Items	% Curve Fitting ± SD		*p*-Value
	CO	OR	
α-helix (1655 cm^−1^)	54.40 ± 5.59	52.23 ± 5.94	0.163
β-sheet (1622, 1627, 1630 cm^−1^)	26.77 ± 2.28	25.44 ± 2.71	0.066
Antiparallel β-sheet (1695 to 1685 cm^−1^)	5.25 ± 6.52	5.08 ± 7.06	0.918
β-turn (1670, 1678, 1680 cm^−1^)	18.82 ± 5.85	22.32 ± 7.41	0.369
α-helix/β-sheet	2.05 ± 0.28	2.07 ± 0.26	0.750

CO = control, OR = organic raising system.

## Data Availability

The data presented in this study are available on request from the corresponding author.

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
