# Peer review of "Biomolecules, Fatty Acids, Meat Quality, and Growth Performance of Slow-Growing Chickens in an Organic Raising System"

_animals, 2022, doi:10.3390/ani12050570_

Round 1

Reviewer 1 Report

The study is evaluating the meat obtained from broilers obtained from a slow-growing breed. The title uses chicken and not broiler.

Organic production requires that all feeding sources were produced under organic certified conditions. The assay presented is either improperly explained and characterized or it is not an organic production system.

As it was described is an hybrid production system:

1) The control group is a conventional (Intensive) production system based on concentrate feeding;

2) The organic group is a conventional (semi-intensive production system, based on the combination of concentrate feeding and grazing.

To stand as an organic production system, the concentrate feeding should be certified as organic (which is not mentioned) and is such situation, both systems were organic.

Beyond this prime defect, some minor problems were identified and appointed in the introduction and material and methods.

We believe that the manuscript should be reviewed, organized by a senior researcher or some one with good English writing and superior conceptual organization.

In my honest opinion it requires major improvements before it stands another review.

Author Response

Response to Reviewer #1

Thank you very much for your comments. We appreciate the time and effort that you have dedicated to providing your valuable feedback on our manuscript. We are grateful to you for your insightful comments on our paper. We have been able to incorporate changes to reflect most of the suggestions provided by you.

Here is a point-by-point response to the reviewer’s comments and concerns. Please find our response (in blue) to reviewer’s comments (in black) below.

Comments and Suggestions for Authors: Reviewer #1

The study is evaluating the meat obtained from broilers obtained from a slow-growing breed. The title uses chicken and not broiler.

We apologize for the confusion. However, Korat chicken (KRC), used in this study, is a slow-growing chicken, not a broiler chicken. The meat quality of this crossbred chicken is distinct from that of commercial broiler chickens. We have therefore decided to use “slow-growing chicken” in the title.

Organic production requires that all feeding sources were produced under organic certified conditions. The assay presented is either improperly explained and characterized or it is not an organic production system.

As it was described is an hybrid production system:

1) The control group is a conventional (Intensive) production system based on concentrate feeding;

2) The organic group is a conventional (semi-intensive production system, based on the combination of concentrate feeding and grazing.

 To stand as an organic production system, the concentrate feeding should be certified as organic (which is not mentioned) and is such situation, both systems were organic.

Beyond this prime defect, some minor problems were identified and appointed in the introduction and material and methods.

We would like to thank for this comment that remind us to explain one of the most important issues. In this study, we would like to compare chickens raise in conventional raising system (CO) and organic raising system (OR) not semi-intensive production system. In the CO group, the chickens were fed with mixed feed derived from commercial feedstuffs, and housed in an indoor pen (5 birds/m2), while those in the OR group, the chickens were raised according to the guidelines of the National Bureau of Agricultural Commodity and Food Standards, the organization that takes care of the organic standard in Thailand. In the OR group, the chickens were fed more than 96% certified organic feed without antibiotics, medicines, chemical fertilizers and pesticides. In addition, the chickens have free access to a pasture area (4 m2/bird) where they can perform their natural behavior.

We believe that the manuscript should be reviewed, organized by a senior researcher or someone with good English writing and superior conceptual organization.

Thank you for your suggestion. This revised manuscript was proven, and edited the English language by editage (https://www.editage.com/). The certificate of English proofreading was attach together with this letter.

In my honest opinion it requires major improvements before it stands another review.

Response to the comments in the manuscript.

  1. Line 50, the word “collagen” is ambiguous

The word ‘collagen’ has been replaced by ‘total collagen’ to avoid ambiguity. (Line 59 and also in abstract, line 33)

  1. Line 53 the words “not all breeds” needs to be rephrased, “not all breeds are adaptable, but the slow growing seem to be suitable for the OR”

The sentence has been rephrased as suggested. (Line 62-63).

  1. Line 63 the sentence “Furthermore, though Southeast Asia is one of the largest chicken-producing countries” incorrect, the Southeast Asia is not a Country.

      The word ‘countries’ has been replaced by ‘regions’. (Line 74)

  1. Line 73, The rate of conversion from ALA into EPA or DHA is relatively low.

The ALA content in pasture is generally low an variable between species

We agree with the reviewer that the conversion rate of ALA to EPA or DHA are relatively low. Moreover, the ALA content in pasture is generally low and varies among species. However, there are studies which have shown that chicken consuming pasture can convert the content of ALA to EPA and accumulate it in their meat (Ponte et al., 2008). It has also been shown that the conversion efficiency of OR chickens is higher than that of CO chicken.

Ponte, P.I.P.; Alves, S.P.; Bessa, R.J.B.; Ferreira, L.M.A.; Gama, L.T.; Bras, J.L.A.; Fontes, C.M.G.A.; Prates, J.A.M. Influence of pasture intake on the fatty acid composition, and cholesterol, tocopherols, and tocotrienols content in meat from free-range broilers. Poult. Sci. 2008, 87, 80-88. http://doi.10.3382/ps.2007-00148

  1. Line 77, I do not understand which are these secondar

The secondary protein structure is the three-dimensional form of local segments of proteins such as α-helix and β-sheet. These secondary protein structure related to nutritive value and protein digestion in animals. The raising system affected lipid oxidation, resulting in a change in secondary protein structure, which was associated with biochemical changes in chicken meat.

  1. Line 78, The phospholipid content and composition are quite constant within the same genetic background

We agree with your opinion. We have deleted the indicated phrase as we found that the information is not essential to understand the background. (Line 88)

  1. Line 99-100, The introduction should not provided the expected results.

We have removed the expected results and modified the sentence accordingly and updated the objective statement of the study. The sentence has been rephrased to “The present study investigates the effects of OR on growth performance, physicochemical properties, and biochemical composition of meat using SR-FTIR in KRC. Furthermore, in this study, KRC was used—mainly to identify its potential as one of the representatives of slow-growing chicken in OR and represent itself.” (Line 109-114)

  1. Line 106, The feed provided to both CO and OR groups is the same?If so, all the items included should be obtained under organic production. Otherwise, the comparison is between intensive and semi-intensive production systems.

As described in response to one of your previous questions, we have described the feed details in the materials and methods section.

  1. Line 134-141, The highlighed text reveals some inconsistency in the text

Thank you for highlighting this. We have corrected them. (Line 154-161)

  1. 150-153, Among liposoluble vitamins (A, D. E and K) they should be presented in IU. The hydrosoluble vitamins should be presenting following the same nomenclature, either B,,, or by their chemical name.

Thank you for highlighting this. We cannot change the unit of K3 from mg to IU because we don’t have any information and we did not make the premix by ourselves; we bought a commercial premix. In addition, we have changed all the pattern of nomenclature of hydrosoluble vitamins. (Line 165-167)

  1. Table 6, The a* coordenate provides values of red (positive) or green (negative), the results are indication that Meat color is greenish

In this study, negative a* was detected in the skin (-0.47) and meat (-0.29) of KRC from the control group, but not in OR. The results are indicating that meat color is greenish. However, this finding is in agreement with Wattanachant et al. [1] who detected negative a* in Thai chicken meat (-0.06). In addition, Fanatico et al. [2] reported that the skin of slow-growing chickens raised indoors was less redness (-0.17) than that of fast-growing chickens and slow-growing chickens raised outdoors. This may indicate that genetics and production system have an influence on meat and skin color of chicken.

[1] Wattanachant, S.; Benjakul, S.; Ledward, D.A. Composition, color, and texture of Thai indigenous and broiler chicken muscles. Poult. Sci. 2004, 83, 123-128.

https://doi.org/10.1093/ps/83.1.123.

[2] Fanatico, A.C.; Pillai, P.B.; Emmert, J.L.; Owens, C.M. Meat quality of slow-and fast-growing chicken genotypes fed low-nutrient or standard diets and raised indoors or with outdoor access. Poult. Sci. 2004, 86, 2245-2255. http://doi.org/10.1093/ps/86.10.2245.

Reviewer 2 Report

This article explores the effects of organic raising systems (OR) on growth performance, meat quality, and physicochemical properties of slow-growing chickens. The idea of the article is novel, and the research content and results are rich, including the comparison of growth performance and meat quality under two groups of feeding systems, and the use of synchrotron radiation-based Fourier transform infrared spectroscopy technology. The following are the specific details:

  1. Please introduce the scientific definition of OR and the difference from free range chicken.
  2. Line 64: countries should be changed to regions.
  3. The contents of fatty acids, total collagen, nucleotides, cholesterol, etc. should be measured and not analyzed in materials and methods.
  4. Line 120-121: The outdoor raising environment in the organic raising system needs to be clearly described in detail, such as the grass outside, how does the grass grow? Will grass be eaten daily by chickens during 21-84 days? Ruzi pasture needs a detailed introduction.
  5. The physical activities were discussed in the article (Line 353-356), but there is no supporting data, such as amount of exercise, outdoor stay time, etc. It will be better to connect them.
  6. What does the organic raising system mentioned in this article refer to? Does it mean that the feed is organic? or is it only raised indoors and outdoors, or both? Need to be described clearly.
  7. Line 134-141: 24 chickens were randomly selected from each group, Were the 24 chickens selected from six pens, 4 birds per pens, 2 males and 2 females? Detailed description is required.
  8. Line 31: The results show that the muscle diameter is not significant (p=0.056), and the description in the abstract is not precise.
  9. In a previous article "Comparison of Carcass Characteristics, Meat Quality, and Blood Parameters of Slow and Fast Grown Female Broiler Chickens Raised in Organic or Conventional Production System", Organic production systems make chickens produce more abdominal fat, this article have the opposite result, please explain.
  10. Line 149: the same format should be here for "d 1 to d 21", "d 22 to 42", and "d 43 to 84" in Table 1.
  11. "Note" is missing in Table 6, please add it.
  12. Line 434: It is recommended to delete KRC.
  13. Line 528: from is changed to between
  14. The breast and thigh meat should be clearly described in the Figure 2, as well as in the results and discussion. Why are the numbers of individuals different in Figures 2A and 2C?

Author Response

Response to Reviewer #2

Thank you very much for your comments. We appreciate the time and effort that you have dedicated to providing your valuable feedback on our manuscript. We are grateful to you for your insightful comments on our paper. We have been able to incorporate changes to reflect most of the suggestions provided by you.

Here is a point-by-point response to the reviewer’s comments and concerns. Please find our response (in blue) to reviewer’s comments (in black) below.

Comments and Suggestions for Authors: Reviewer #2

This article explores the effects of organic raising systems (OR) on growth performance, meat quality, and physicochemical properties of slow-growing chickens. The idea of the article is novel, and the research content and results are rich, including the comparison of growth performance and meat quality under two groups of feeding systems, and the use of synchrotron radiation-based Fourier transform infrared spectroscopy technology. The following are the specific details:

  1. Please introduce the scientific definition of OR and the difference from free range chicken.

Thank you for highlighting this. Accordingly, we have updated the introduction section of the revised manuscript. “The organic raising system (OR) is a poultry management system where the birds are fed only with organic feed (produced without chemical fertilizers and pesticides) and allowed to grow and express their natural behaviors without the use of chemicals like antibiotics and other drugs [3]. All OR are free-range systems, as they allow free outdoor access; however, the reverse is not true, as the free-range systems that are not OR use general feed, medications and chemicals [4]”. (Line 51-56)

  1. Line 64: countries should be changed to regions.

The correction has been made. (Line 74)

  1. The contents of fatty acids, total collagen, nucleotides, cholesterol, etc. should be measured and not analyzed in materials and methods.

The correction has been made. (Line 204, 217, 227, 240)

  1. Line 120-121: The outdoor raising environment in the organic raising system needs to be clearly described in detail, such as the grass outside, how does the grass grow? Will grass be eaten daily by chickens during 21-84 days? Ruzi pasture needs a detailed introduction.

Thank you for your meticulous review. We have updated the detail about outdoor raising environment in the organic raising system and explained how the Ruzi pasture was maintained before and during the experiment. (Line 138-143)

  1. The physical activities were discussed in the article (Line 353-356), but there is no supporting data, such as amount of exercise, outdoor stay time, etc. It will be better to connect them.

We have added some information about the period (daily) time of outside activities of the chickens. These activities are related to altered protein content in chicken meat. (Line 400-403)

  1. What does the organic raising system mentioned in this article refer to? Does it mean that the feed is organic? or is it only raised indoors and outdoors, or both? Need to be described clearly.

The organic raising system in this study refers to a poultry management system in which birds are fed more than 96% organic feed without antibiotics, drugs, chemical fertilizers, and pesticides, and have free access to outdoor pasture (4 m2/bird) to perform their natural behaviors. The chickens in organic group were raised according to the guideline of National Bureau of Agricultural Commodity and Food Standards in Thailand. This information has been added in the part of “Materials and Methods”. (Line 127-128, 138-143)

  1. Line 134-141: 24 chickens were randomly selected from each group, Were the 24 chickens selected from six pens, 4 birds per pens, 2 males and 2 females? Detailed description is required.

Thank you for pointing this out. Accordingly, we have updated the information in the materials and methods section. “At slaughter age (d 84), 24 chickens from each group were randomly selected and electrically stunned, and their feathers were removed with a machine. Then they were scalded and eviscerated manually. The carcass composition and meat quality were measured in 12 chickens per treatment (6 males and 6 females). The proximate composition, FA profile, cholesterol content, and nucleotide content were estimated from the breast and thigh meat samples obtained from the remaining 12 chickens (6 males and 6 females). The samples were stored at −20 °C till analyses”. (Line 155-161)

  1. Line 31: The results show that the muscle diameter is not significant (p=0.056), and the description in the abstract is not precise.

The word “muscle diameter” has been removed from the abstract. (Line 33)

  1. In a previous article "Comparison of Carcass Characteristics, Meat Quality, and Blood Parameters of Slow and Fast Grown Female Broiler Chickens Raised in Organic or Conventional Production System", Organic production systems make chickens produce more abdominal fat, this article have the opposite result, please explain.

Thank you for your suggestion. Accordingly, we have updated the information and revised the sentence to make it more clearly in the results section. (Line 345-353) “In contrast, Comert et al. [9] have demonstrated a higher amount of abdominal fat in chicken grown in the OR system than those grown in the CO system. This difference could be attributed to the different genotypes and the sex of birds used in the two studies. Taken together, we inferred that the OR chickens are exposed to increased physical activity than the CO chickens, which, though increases the energy metabolism rate and reduces the abdominal fat accumulation, is not sufficient enough to increase the carcass yield. Furthermore, sex and genotype are the other important factors affecting the carcass characteristics, therefore, should be carefully considered [9,56].”

  1. Line 149: the same format should be here for "d 1 to d 21", "d 22 to 42", and "d 43 to 84" in Table 1.

The correction has been made. (Table 1)

  1. "Note" is missing in Table 6, please add it.

We have updated the footnotes under Table 6. (Line 435-436)

  1. Line 434: It is recommended to delete KRC.

The word “KRC” has been deleted. (Line 481)

  1. Line 528: from is changed to between

The correction has been made. (Line 576)

  1. The breast and thigh meat should be clearly described in the Figure 2, as well as in the results and discussion. Why are the numbers of individuals different in Figures 2A and 2C?

We have revised the sentence and added the word to clarify in figure 2 (Line 765-767), results and discussion. (Line 603, 607, 610, 616)

About the numbers of individuals different in Figures 2A and 2C we would explain that we analyzed a total of 12 chickens (1 male and 1 female from each replicate of each treatment group). To generate the figure, we used the average of the male and female values for each replication. Figure 2A explained the meat quality parameters and biochemical compounds measured by the SR-FTIR technique, whereas Figure 2C describes the fatty acids in chicken meat measured by gas chromatography technique. In addition, in Figure 2A, we used only breast tissue. Thigh meat contains different muscle fiber type and is heterogeneous, which is not suitable to analyze using the SR-FTIR technique. We generate this figure using the 6 breast tissues data obtained from 12 chickens. On the other hand, in Figure 2C, we analyzed the fatty acids composition of breast and thigh meat. However, we could not measure fatty acid data for both breast and thigh tissue from one chicken in each treatment although we measured several times. Thus, we generated the figure from the data obtained from 10 chickens in Figure 2C.

Reviewer 3 Report

Line 108-109: provide the directions of the world to the coordinates and the information that temperature and humidity relate to air. From what source was the information on climatic conditions obtained?

In Conclusion, some statements do not come directly from the analysis results. They are only assumptions. 

Author Response

Response to Reviewer #3

Thank you very much for your comments. We appreciate the time and effort that you have dedicated to providing your valuable feedback on our manuscript. We are grateful to you for your insightful comments on our paper. We have been able to incorporate changes to reflect most of the suggestions provided by you.

Here is a point-by-point response to the reviewer’s comments and concerns. Please find our response (in blue) to reviewer’s comments (in black) below.

Comments and Suggestions for Authors: Reviewer #3

Line 108-109: provide the directions of the world to the coordinates and the information that temperature and humidity relate to air. From what source was the information on climatic conditions obtained?

The information about the directions of the world to the coordinates has been changed from “The experimental site was located at coordinates 14.889868, 102.0048......” to “The experimental site was located at coordinates latitude runs 14°53'13'' North, Longitude runs 101°59'42'' East.” (Line 122-124), and “Nakhon Ratchasima Meteorological Department, Thailand” has been added as a source of information for climatic conditions. (Line 125)

In Conclusion, some statements do not come directly from the analysis results. They are only assumptions. 

Thank you for your suggestion. Accordingly, we have updated the conclusion section. The conclusion has been rephrased to “In conclusion, our results reveal that OR has no negative effect on the growth performance of slow-growing chickens. The study show that the OR system has positive effect on the meat characteristics, especially meat color and texture, biochemical compounds such as proteins (amide I and amide II), total collagen (amide III), and beneficial FA (PUFA, DHA, and ALA), which determine the nutritional value of meat. The findings of this study demonstrate the potential of OR for commercial adoption in tropical regions like Southeast Asia. Furthermore, the study demonstrates the efficiency of SR-FTIR to determine the differences in the biochemical compounds, which could serve as markers to monitor meat quality traits. Collectively, these findings provide insights into the relative roles of raising systems in KRC chickens and can help producer to produce nutritionally rich quality products while maintaining animal welfare standards”. (Line 770-780)